# Shift in vacuolar to cytosolic regime of infecting *Salmonella* from a dual proteome perspective

Ursula Fels[1,2]☯, Patrick Willems[1]☯¤, Margaux De Meyer[1,2], Kris Gevaert[2,3], Petra Van Damme[1]☯ *

1 iRIP unit, Laboratory of Microbiology, Department of Biochemistry and Microbiology, Ghent University, Ghent, Belgium, 2 VIB-UGent Center for Medical Biotechnology, Ghent, Belgium, 3 Department of Biomolecular Medicine, Ghent University, Ghent, Belgium

☯ These authors contributed equally to this work.
¤ Current address: VIB-UGent Center for Plant Systems Biology, Ghent, Belgium. Department of Plant Biotechnology and Bioinformatics, Ghent University, Ghent, Belgium.
* Petra.VanDamme@ugent.be

**Data Availability Statement:** The mass spectrometry proteomics data have been deposited to the ProteomeXchange Consortium via the PRIDE [81] partner repository with the dataset

## Abstract

By applying dual proteome profiling to *Salmonella enterica* serovar Typhimurium (*S*. Typhimurium) encounters with its epithelial host (here, *S*. Typhimurium infected human HeLa cells), a detailed interdependent and holistic proteomic perspective on host-pathogen interactions over the time course of infection was obtained. Data-independent acquisition (DIA)-based proteomics was found to outperform data-dependent acquisition (DDA) workflows, especially in identifying the downregulated bacterial proteome response during infection progression by permitting quantification of low abundant bacterial proteins at early times of infection when bacterial infection load is low. *S*. Typhimurium invasion and replication specific proteomic signatures in epithelial cells revealed interdependent host/pathogen specific responses besides pointing to putative novel infection markers and signalling responses, including regulated host proteins associated with *Salmonella*-modified membranes.

## Author summary

As causative agents of infectious diseases, intracellular bacterial pathogens have evolved diverse immune escape and survival strategies while thriving inside their hosts. Whereas the residence of the pathogen inside the host can impact the functioning of the latter drastically, the bacterial macromolecular content is typically vastly outstripped by that of the host. This sizeable difference in macromolecular content has thus far hindered proteome profiling of bacteria and its infected host in an integrated and unbiased manner. Here, the power of dual proteome profiling—a universal mass spectrometry-based proteomic strategy for unravelling the intricate interplay between pathogen and host—is presented for the exploration of host as well as pathogen proteomic responses over the time course of the infection, thereby advancing our understanding of infectious diseases.

identifier PXD018610 for DDA data and with the dataset identifier PXD019863 for DIA data. The generated spectral libraries, FASTA databases, and search results are organized in the Open Science Framework (OSF) project page [82] and are available at https://osf.io/u96bk/.

**Funding:** This work was supported by the European Research Council (ERC) under the European Union's Horizon 2020 research and innovation program (PROPHECY grant agreement No 803972) to PVD, the Fonds Wetenschappelijk Onderzoek (FWO-Vlaanderen) (project number G051120N) to PVD, and by Ghent University Concerted Research Actions (grant BOF23/GOA/001) to PVD and (grant BOF14/GOA/013) to KG. PW and MDM authors received a salary from the PROPHECY grant. The funders had no role in study design, data collection and analysis, decision to publish, or preparation of the manuscript.

**Competing interests:** The authors have declared that no competing interests exist.

## Introduction

*Salmonella enterica* serovar Typhimurium (*S.* Typhimurium) is an enteric facultative intracellular pathogen with wide-ranging disease outcomes grading from self-limiting gastroenteritis to systemic infection. As a foodborne pathogen, *S.* Typhimurium invades the intestinal tissue via different routes through infection of a diverse range of host cells, including epithelial cells (e.g., M cells and absorptive epithelial cells) and immune cells (e.g., dendritic cells and macrophages), in which they proliferate [1, 2]. Infection by *Salmonella* is mediated by its two type III secretion systems (T3SSs); T3SS-1 and -2, encoded within *Salmonella* pathogenicity island 1 (SPI-1) and -2 (SPI-2), respectively [3]. Along with the T3SS-1 components, the 40-kb SPI-1 gene cluster encodes chaperones, type III effector proteins (T3Es) and transcriptional regulators that modulate expression of many virulence genes within, but also outside SPI-1 [4]. Whereas T3SS-1 and its effectors are mainly associated with the invasion of non-phagocytic epithelial cells [5], T3SS-2 delivers effectors that promote *Salmonella* replication inside phagocytic as well as non-phagocytic cells [6–8]. To invade epithelial cells, *Salmonella* delivers through its T3SS-1 a first batch of effectors into the host cytoplasm to provoke actin cytoskeleton rearrangements that allow for bacterial uptake by macropinocytosis. More specifically, lamellipodia and filopodia-containing protrusions, called membrane ruffles, emerge and enclose the pathogen [9, 10]. This active invasion of epithelial cells mediated by T3SS-1 and its cognate effectors is thought to play a key role in gut inflammation during salmonellosis [1, 5]. A first set of pro-inflammatory responses (e.g., NF-κB signalling) are detected upon entry of the pathogen owed to both manipulation of cell death pathways and/or tight junctions via effectors, and by recognition of peptidoglycan, flagellin or lipopolysaccharide by host pattern recognition receptors (PRRs) [11]. Following internalization, *Salmonella* resides inside a phagosome-like compartment, dubbed the *Salmonella*-containing vacuole (SCV), where it further rewires host cell processes to create a replicative niche mainly through the action of T3SS-2 effectors [12]. In essence, T3SS-2 effectors maintain the SCV membrane, navigate juxtanuclear SCV positioning and elicit filamentous membrane extensions from the SCV along microtubules, termed *Salmonella*-induced filaments (SIFs) [13]. SIFs have been observed in various cell types and exhibit endosomal markers like LAMP1, Rab7 and vacuolar ATPase [13]. Interestingly, the onset of SIF biogenesis at 4–8 hours post-infection coincides with intercellular bacterial replication and has therefore been postulated to facilitate nutrient acquisition from the host, amongst others [14–17]. Remarkably, in about 20% of infected epithelial cells, bacterial replication also occurs outside the SCV in the cytosol where flagellated bacteria hyper-replicate and account for the majority of net replication [18]. Moreover, some SCVs are displaced from their juxtanuclear positioning in the host cell towards the cell periphery [19]. Both phenomena, i.e., vacuolar escape and centrifugal SCV movement, have been linked to dissemination of the pathogen to other cells to repeat the infectious cycle [19]. Moreover, the various (subcellular) environments encountered by these bacterial subpopulations during infection further lie at the basis of the observed heterogeneity of bacterial subpopulations *in vitro* [20] and *in vivo* [21]. Specific bacterial subpopulations were for example found to reprogram infected hosts to promote long-term bacterial survival [22].

As such, it is clear that over the course of an infection, an intricate and complex interplay between *Salmonella* and its host occurs that ultimately defines the infection outcome. As diverse successive environmental cues shape both the host and bacterial proteome, this highlights the importance of studying both proteomes in an integrative manner to unwire this multifaceted interaction and to obtain holistic insights. The characterization of molecular mechanisms underlying interactions between pathogen and host has extensively advanced over the last two decades, mainly through omics-informed infection studies in cellular as well as animal infection models.

Staples *et al.* [22] characterized non-growing, metabolically active subsets of *Salmonella* associated with persistence [23, 24] by dual RNA sequencing (RNA-seq). This simultaneous analysis of the transcriptomes of the host and the pathogen revealed that this specific bacterial subpopulation secrete effectors implicated in reprograming macrophages to promote anti-inflammatory macrophage polarization linked with bacterial long-term survival [25]. While the sensitivity of transcriptome analysis outperforms proteome studies, especially in the context of infection, the lower growth rates typically associated with infection [26], bacterial virulence [27, 28] and persistence, are associated with a considerably higher gene expression noise [29] and extensive post-transcriptional regulatory control. Therefore, mRNA levels in bacterial populations with reduced growth rates often correlate poorly with protein levels, stressing the need for protein-based studies. However, as the theoretical *Salmonella* proteome (~4,500 proteins) is vastly underrepresented in the context of the infected host (~20,000 proteins), especially when considering an up to ~1500 fold difference in the *Salmonella* versus human cellular total proteome content in the context of infection [30], studying both proteomes simultaneously and interactively entails proven technical challenges [30, 31] that are further complicated by the considerable heterogeneity in bacterial and host subpopulations as mentioned before.

Accordingly, proteome profiling of *Salmonella* and its infected host has thus far mainly focused on studying the reprogramming and signalling responses of the host or bacterial adaptive responses disjointedly [32–38], or alternatively, studied the interactions between host and pathogen to provide insights into the complex network of effector/host interactions during infection [39].

By significantly improving peptide detection and quantification, we recently reported on the hybrid use of data-dependent and data-independent acquisition spectral libraries to empower what we called dual proteome profiling [31]. In our current study, we used this optimized workflow to perform time-dependent dual proteome profiling of infection-relevant *S.* Typhimurium pathogen encounters with its host, without the need of *a priori* bacterial pathogen enrichment. More specifically, through the use of an intracellular epithelial infection model (i.e., *S.* Typhimurium infected HeLa cells), dual proteome profiling reports for the first time on the extensive protein-level dynamics during *Salmonella* infection thereby unveiling a detailed and integrative proteomic perspective on host-pathogen interactions over the time course of infection. This way, the regulation of host proteins associated with *Salmonella*-modified membranes and/or the formation of SCVs or SIFs was revealed besides the induction of epithelial cell differentiation in response to *Salmonella* infection. The later potentially representing a pathogenic strategy of intracellular bacteria to reprogram the host cell thereby enabling survival and pathogenic spread [25].

## Results

### Epithelial salmonella infection model

In line with previous reports in HeLa cells, while at lower multiplicities of infections (MOI or ratio of bacteria to cells of 10) only ~5% of HeLa cells were infected and since at modest MOI increases, higher bacterial loads rather than an increase in the overall number of infected cells has been reported due to *Salmonella* cooperative entry, an MOI of 100 resulted in the infection of over half of the cell population [18, 40–42] and was opted for dual proteome profiling (see below results). As bacterial load has proven to impact host signalling and host responses [2, 18, 20, 42], we assessed host cell cytotoxicity and host cell counts in function of bacterial load over the time course of the infection in real-time (S1 Fig). In control cultures (non-infected cells and cells infected with a non-invasive *ΔprgH* mutant strain [43]) as well as cultures infected at

low MOI (MOI 10), no significant impact on cell viability and proliferation was apparent (i.e., inferred from the expected ~2-fold increase in cell counts over the time course of measurement (24 h) considering the 28.2 h doubling time of HeLa CCL2 cells [44]). At an MOI of 100, host cell counts and viability was not significantly affected up till 10 hpi when compared to the control setups (S1A and S1B Fig), while at 24 hpi a 16% increase in cytotoxicity was observed (i.e., when considering maximum cytotoxicity (total lysate reference) compared to the control setups)(S1C and S1D Fig). Using a constitutively eGFP-expressing SL1344 strain [45], bacterial load could be inferred from the total fluorescence (RFU) and green object counts. Here, from the two-step progression curve observed, a first steep rise in fluorescence–and total green object counts–was observed between 2 and 5 hpi, followed by a plateauing phase and a less steep increase at ~8 hpi which plateaued again at ~16–17 hpi (S1E Fig). This second phase coincides with the occurrence of hyper-replicating *Salmonella* in the cytosol [46]. Further, green object sizes also indicated that a maximum green object size was observed at ~8 hpi before a ~2-fold decrease in size (S1F Fig), both phases aligning with the accumulation of *Salmonella* in SCVs before vacuolar escape and more diffuse bacterial spread in the cytosol, respectively [46].

## Dual proteome profiling of salmonella-infected epithelial cells

Proteome samples of *S*. Typhimurium infected human HeLa cells were prepared to monitor differences in steady-state protein expression levels at 2, 4, 8, 16 and 24 hpi in biological quadruplicates. Following trypsin digestion, each sample was analysed by LC-MS/MS in both DDA and DIA mode and data analysed respectively using MaxQuant [47] against a composite database containing *Salmonella* and human UniProtKB protein entries, and EncyclopeDIA [48] using our optimized hybrid spectral library workflow [31]. The latter combines experimental libraries with predicted spectral libraries to optimize the search space for identifications, shown to facilitate dual proteome profiling [31] and suited for DIA-based exploration of *S*. Typhimurium infected epithelial host cells.

When considering all samples analysed in both DDA and DIA acquisition modes, a total of 7,821 proteins were identified, comprising 6,830 human proteins (87% of all protein identifications) and 991 *S*. Typhimurium (13% of all protein identifications and 21% of the annotated *S*. Typhimurium proteome) proteins.

In a bit more detail, by DDA, 6,696 unique proteins (i.e., 767 *S*. Typhimurium and 5,929 human proteins) (S1 Table) and, by DIA, 6,734 unique DIA proteins (i.e., 853 *S*. Typhimurium and 5,881 human proteins) (S2 Table) were identified, implying that the number of human proteins identified remained by and large unaffected when comparing DDA versus DIA data, while a notable increase in *S*. Typhimurium proteins identified (~10%) was observed (Fig 1A). The latter is further accentuated when considering the 1.6-fold increase in the unique number of *Salmonella* peptides identified (i.e., 3,851 versus 6,306 unique peptides by DDA and DIA, respectively) (S1 and S2 Tables).

From the host perspective, global or individual principal component analysis (PCA) revealed a clear variability between control versus *S*. Typhimurium infected HeLa cells (component 1) as well as a temporal response (component 2) where proteomes from earlier timepoints post infection (2, 4 and 8 hpi) can clearly be distinguished from later timepoints (16 and 24 hpi) (Figs 1B, 1C, S2A and S2B). This temporal response is also reflected in the *Salmonella* recordings (component 1, 53% for DDA and 34.1% for DIA data) (S2C and S2D Fig).

To monitor differences in protein levels between all DIA and DDA setups and replicates analysed, multiscatter plots of protein intensity values (log2) were generated. When grouping replicate samples per setup, the average intra-setup Pearson correlations appeared higher for

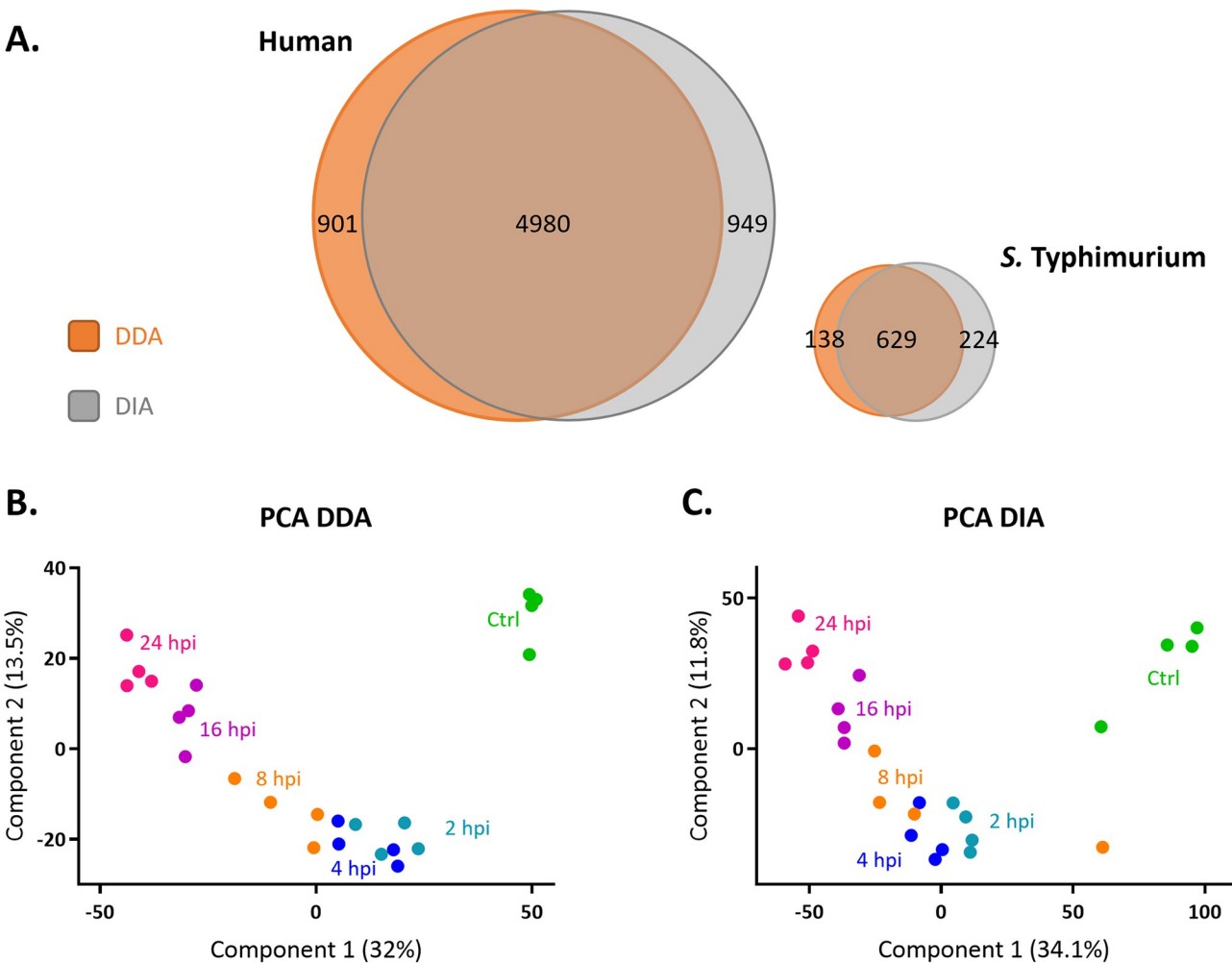

**Fig 1. DDA and DIA dual proteome profiling of *Salmonella*-infected epithelial cells. (A)** Venn diagram created using Venny (http://bioinfogp.cnb.csic.es/tools/venny/index.html) showing the total number of human (left Venn diagram) and *S.* Typhimurium (right) proteins identified using DDA (orange) and DIA (grey) data (both at FDR ≤ 0.01). Variability of DDA **(B)** and DIA **(C)** replicate samples represented in PCA plots. Green, cyan, blue, orange, purple and pink circles represent control (Ctrl), 2, 4, 8, 16 and 24 hpi samples, respectively.

DDA versus DIA (average and minimum correlations of 97% for DDA samples, and average and minimum correlations of 95% and 92%, respectively for DIA samples) with correlations ranging between 92% and 98% for all DDA datasets, and between 84% and 97% for DIA datasets when considering average correlations among all different setups (inter-setup) analysed (see correlation plots in S3 and S4 Figs).

Further, considering the protein expression ranges observed, a dynamic range spanning over 5 orders of magnitude (max. ~365,000-fold change) for DIA data, while only over 4 orders for DDA data was detected (max. ~57,000-fold change) (S1, S2, S3 and S4 Tables).

## Improved sensitivity of DIA to study Salmonella protein downregulation during infection

After filtering for valid values (minimum 3 valid values in at least one group/setup), median normalized expression per species and imputation of missing values, human and *S.* Typhimurium protein expression values were compared in a pairwise manner with the HeLa control and

the *S.* Typhimurium proteomes at 2 hpi, respectively. This selection left 4,666 human (S3 Table) and 479 *S.* Typhimurium (S4 Table) protein groups for comparison in case of the DDA data, and 5,818 human (S5 Table) and 844 *S.* Typhimurium (S6 Table) protein groups in case of the DIA data. In line with our dual proteome profiling results reported using artificial proteome mixtures [31], and while the *S.* Typhimurium protein identification rates in DIA only modestly increased (767 vs. 853 *S.* Typhimurium protein identifications), the number of reliably quantified *S.* Typhimurium proteins drastically increased with 76% (1.8-fold) when using DIA data (479 vs. 844 quantified proteins). Further, from the differential expression analysis, it is clear from the corresponding profile plots that both up- as well down-regulation (compared to 2 hpi) of *S.* Typhimurium proteins can comprehensively be studied with DIA data, while expression analysis is mostly confined to upregulated *S.* Typhimurium proteins with DDA data (S5 Fig). This discrepancy can be explained by the limited DDA over DIA sensitivity and the low *S.* Typhimurium protein content in the proteome samples of infected cells, especially at the earliest time points post infection.

Consequently, studying DDA-based *S.* Typhimurium protein downregulation during infection is rather limited to only a few [8] highly expressed *S.* Typhimurium proteins already identified and quantified at 2 hpi. Because of this finding, we further focussed our dual proteome expression analysis to DIA data. Nonetheless, generally similar trends and overlapping regulations could be observed among the DDA and DIA data (S1, S3 and S4 Tables). The intensities of significantly regulated proteins after multiple sample ANOVA testing (FDR ≤ 0.01 and an S0 of 0.1) are shown as heat maps and/or profile plots for human and *S.* Typhimurium regulated proteins, respectively (S5 and S6 Figs, and corresponding S5 and S6 Tables). Overall, 1,742 human and 436 *S.* Typhimurium protein groups showed a significantly regulated temporal response in DIA data during the time-course of infection (FDR ≤ 0.01), with proteins showing similar time-specific responses indicated in assigned clusters (S5 and S6 Figs).

## Host cell responses to Salmonella infection reveal induction of host immunity and epithelial cell differentiation

To map the epithelial host cell response to *S.* Typhimurium infection over time, a gene ontology (GO) enrichment analysis (GOBP (Biological Process), GOCC (Cellular Component), GOMF (Molecular Function), general GO and UniProt keywords (FDR 0.02)) of the significantly regulated proteins (FDR 0.02) identified in the DIA-MS analysis was performed (Fig 2 and S7 Table).

At early time points post-infection (i.e., 2 hpi), interleukin-1 mediated signalling and protein polyubiquitination are upregulated indicative of the induction of host immunity, while intriguingly, at late time points post infection, cell adhesion, extracellular matrix organizations and related terms, e.g., 'collagen-containing extracellular matrix' (e.g., including collagen α-3, thrombospondin-1 and Galectin-1) [50] and 'cell adhesion mediated by integrin' (e.g., with up to over 7-fold upregulation of the integrins α-2 (ITGA2), α-5 (ITGA5), α-V and β, besides ICAM-1, ADAM9 and vitronectin)) represent upregulated processes opposed to intracellular actin cytoskeleton downregulation. Some selected representative protein profile plots of the aforementioned significantly regulated categories are shown in Fig 3. While the marked induction of host immunity, host integrin signalling and extracellular matrix organization is in line with previous proteomic findings of infected epithelial cells [32], we are the first to theorize that jointly, these cellular and phenotypic hallmarks reflect epithelial cellular differentiation induced early upon infection [51].

Overall, besides demonstrating a specific and complex response to bacterial infection over time, our findings seem to indicate that cell differentiation may generate physiological

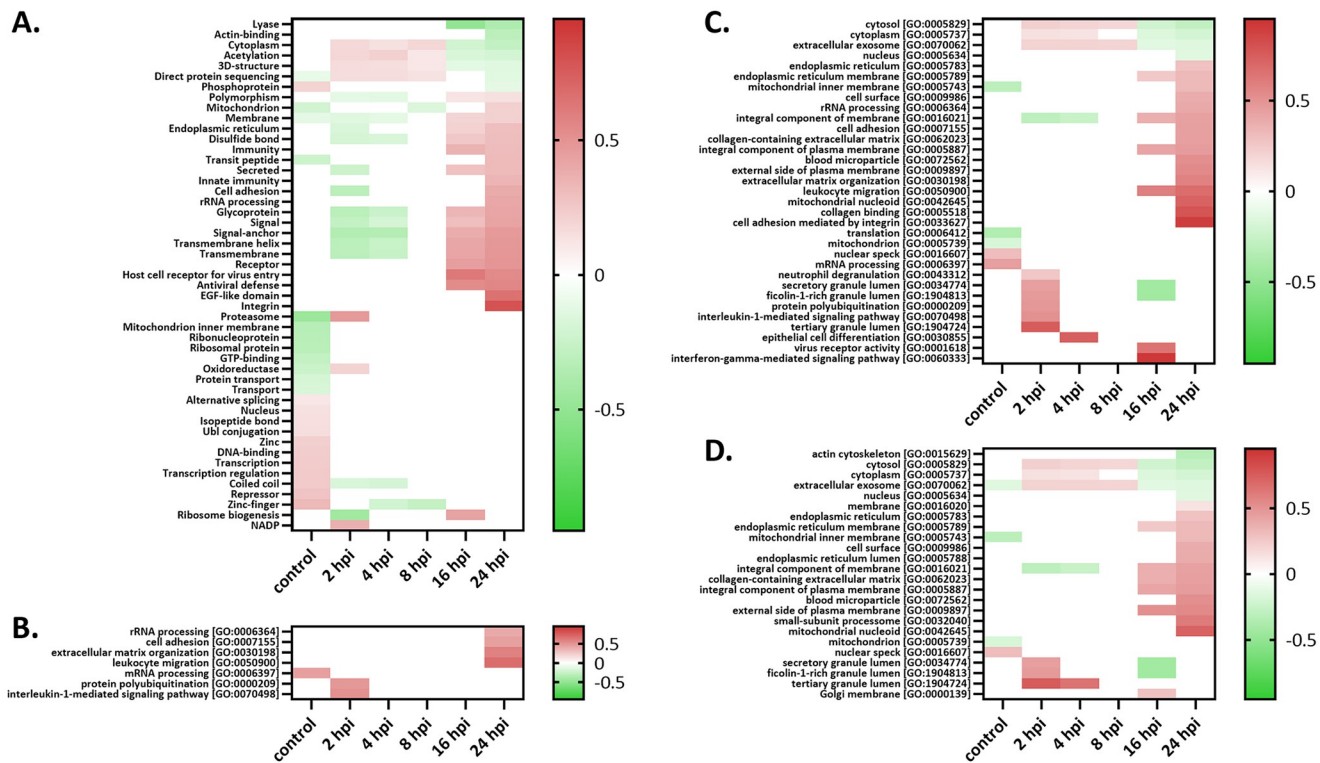

**Fig 2. Heatmaps of significantly regulated (FDR ≤ 0.02) normalized human protein annotation enrichment scores across all six conditions (averaged expression values) assayed by data-independent acquisition (DIA).** Term enrichment was determined using the 1D annotation enrichment algorithm embedded in the Perseus software suite and p-values were corrected for multiple hypotheses testing using the Benjamini and Hochberg false discovery rate. Enriched annotation terms (FDR ≤ 0.02) in the categories **(A)** UniProt keywords, **(B)** GOBP (Biological Process), **(C)** general GO and **(D)** GOCC (Cellular Component) are shown as heatmaps and colour coded according to the 1D enrichment scores calculated [49], with red and green colouring indicating enriched and depleted annotations, respectively.

heterogeneity in *S.* Typhimurium infected epithelial cells as previously shown for the *S.* Typhimurium mediated reprogramming of macrophage hosts towards alternative anti-inflammatory (i.e., M2) polarized macrophages, a pathogenic strategy enabling (a subpopulation of) intracellular bacteria to survive and reprogram their host cell to enable bacterial survival and pathogenic spread [25].

## Regulated expression of Salmonella-modified membranes host proteins implicated in formation of SCVs or SIFs

Since the creation of an extensive membrane network within its host lies at the basis of *Salmonella*'s infection strategy, we turned to a study reporting on a comparative proteome analysis of *S.* Typhimurium infected HeLa cells to obtain more detailed insight into expression changes of the proteome of internal membrane systems [52]. More specifically, Reuter et al. identified host proteins associated with *Salmonella*-modified membranes (SMM) shown to be enriched for transport proteins, GTPases, cytoskeleton and membrane linker proteins as well as proteins implicated in clathrin and coatomer-mediated transport [52]. From the high-confidence set of host proteins found associated with SMM (247 proteins), 213 proteins (86%) were also identified in our DIA analyses. Interestingly, of these, 84 (39%) were found to be significantly regulated after multiple sample ANOVA testing (FDR ≤ 0.01 and an S0 of 0.1) (see columns 'SMM

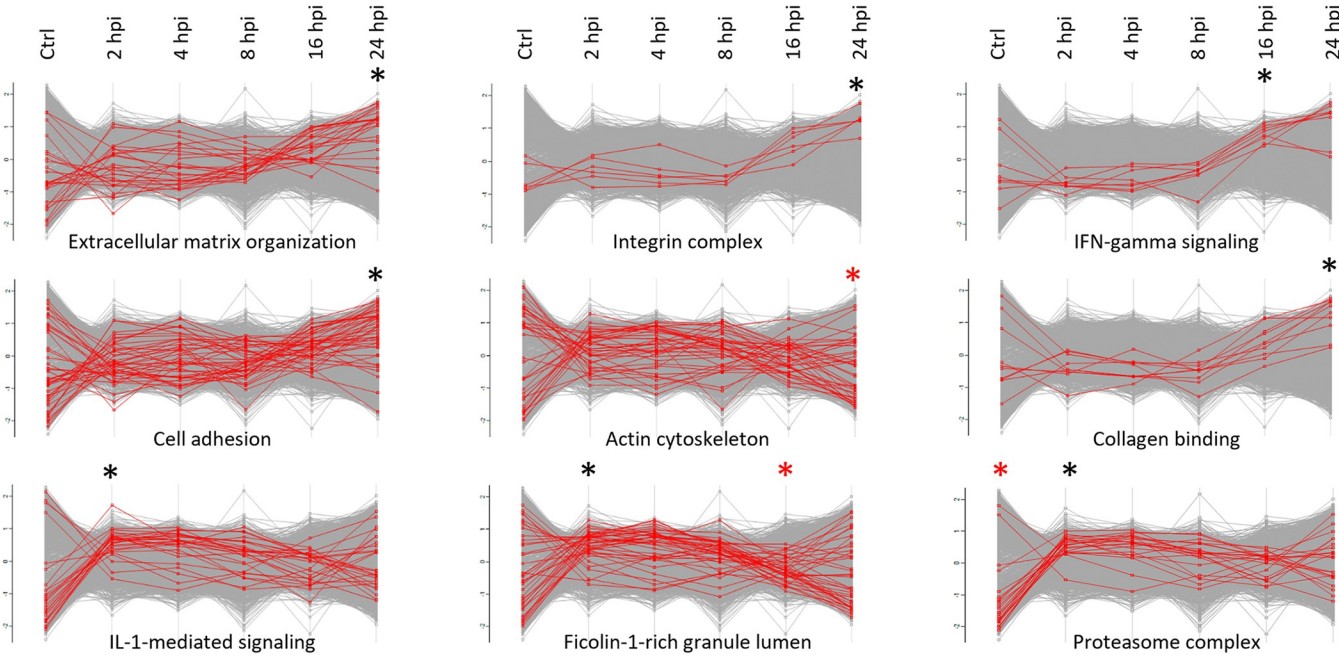

**Fig 3. Protein profile plots of significantly regulated (FDR ≤ 0.02) normalized human annotation enrichment scores across all 6 conditions (averaged expression values) assayed by data-independent acquisition (DIA).** Term enrichment was determined using the 1D annotation enrichment algorithm embedded in the Perseus software suite and p-values were corrected for multiple hypotheses testing using the Benjamini and Hochberg false discovery rate. Only corrected p-values ≤ 0.02 were considered. Selected GO terms and keyword annotations for with significantly altered protein abundancies were observed (i.e., extracellular matrix organization (GOBP, #26); upregulation at 24 hpi, cell adhesion (GOBP, #45); upregulation at 24 hpi, IL1-mediated signalling (GOBP, #27); upregulation at 2 hpi, integrin complex (keyword, #6); upregulation 24 hpi, actin cytoskeleton (GOCC, #26); downregulation 24 hpi, Ficolin-1-rich granule lumen (GOCC, #40); upregulation 2 hpi and downregulation 16 hpi, IFN-gamma signalling (GO, #8); upregulation 16 hpi, collagen binding (GO, #10); upregulation 24 hpi, proteasome complex (keyword, #22); down in control and upregulation 2 hpi) are shown for all setups analysed. Red and black asterisk above the profile plots means significant downregulation and upregulation of the corresponding proteins in the annotation group, respectively.

proteome (PMID: 25348832)' and 'ANOVA regulated SMM protein' S5 Table). By itself, regulation of SMM protein expression over the time-course of infection was found to be highly statistically significant as determined by a chi-square test of independency (p < 0.01). Regulated SMM proteins included among others transport proteins (Phosphate carrier protein MPCP, Translational endoplasmic reticulum ATPase (VCP), ATP synthases F(0) complex subunits 1 and 0), proteins implicated in clathrin/vesicle-mediated transport (AP2A1, AP2B1 and TMED9), cytoskeleton and membrane linker proteins (Ezrin, Plastin-3, Myosin I, the Arp2/3 complex and Catenin alpha-1) besides several GTPases (including the over 10-fold upregulation of the Ras-related protein Ral-A). In line, annotation enrichment analysis showed significantly reduced protein abundancies of SMM proteins in the control setup, pointing to an overall general increase in SMM proteins upon *Salmonella* infection (p-value < 0.01). It is noteworthy that these regulated SMM targets hold several components of host pathways previously shown to be implicated in formation of SCVs or SIFs (e.g., endosomal and recycling pathways). These regulated targets include the regulated small Rab GTPases Rab7a [53], Rap-1b [54] and Rab10 [52]–all with known fundamental roles in cellular membrane dynamics besides representing hub targets of bacterial effector proteins [55]–next to the observed regulations of the AP-2 complex subunit β (AP2B1) [54], the F-actin binding protein utrophin [52], and the cytoskeleton(-linked) proteins desmoplakin [56] and Filamin [57]. Overall, our data demonstrates that a multitude of infection-relevant host proteins from the SMM are regulated in expression over the time course of infection, thereby representing interesting targets for further follow-up studies. The 29 ANOVA regulated SMM proteins (from the 84 regulated SMM

proteins identified in total) signifying GTPases or proteins implicated in transport or cytoskeleton and membrane organizations (selected form keywords) are listed in S8 Table. In conclusion, in the context of infection, our work reports the regulation of host proteins associated with *Salmonella*-modified membranes including targets implicated in formation of SCVs or SIFs.

## Temporal Salmonella proteome adaptations to adjust to the cytosolic lifestyle in infected epithelial cells

Reflecting the virulence lifestyle of infecting *Salmonella*, the pronounced activity of the two-component system *phoPQ* known to be exerted at later time-points post infection was evidenced by the upregulation of its two components PhoP and PhoQ at later time points and its downstream impact on SPI-1 and SPI-2 expression more generally [58] (Fig 4). More specifically, an enrichment of *ΔphoPQ* regulon targets [59] (i.e., a group of genes regulated as a unit, here genes regulated upon deletion of *phoPQ*) found upregulated (i.e, SipC, SopB, SipB, InvB, PrgH, InvG, PrgI) or downregulated (SodCa, PagC, PhoN, WrbA, UgpB, PdgL, AroQ, PotD, Rna, MsrA, YdcW, SseJ, Tal, DkgA, YdcR, PhnO, SseL) was observed at early (2 hpi) and late (16–24 hpi) times post-infection, respectively (FDR $\leq$ 0.05) (S6 Table and Fig 4A). These time-dependent PhoPQ dependencies are in line with the well-established positive regulation of PhoPQ on SPI-2 expression. Other temporal *Salmonella* proteome adaptations were indicative of other affected regulons, with an overall positive regulation of SPI-1 (e.g. *hilA*, *hilC*, *fliZ*) or SPI-2 (e.g. *ssrA*, *slyA*) at early and later time points of the infection, respectively (Fig 4).

Further, indicative of the metal ion shortage encountered by the bacterial pathogen when residing in the host [59, 60], significant upregulation of metal ion transporters and uptake systems for zinc (ZnuA, up to over 3-fold upregulation at 24 hpi), iron (IroBN ($>$ 8-fold), EntBF ($>$ 6-fold), SitAB ($>$ 17-fold), Fur ($>$ 13-fold), Mrp ($>$ 2-fold), FepAB ($>$ 4-fold) and CirA ($>$6-fold)) and manganese (SitAB) was observed (S6 Table). Also, expression of the ferric uptake regulator Fur—which binds to promoters of target genes implicated in iron acquisition and transport in $Fe^{2+}$-rich conditions thereby preventing target transcription–was lowered (Fig 4B). In accordance, a significant impact on the *fur* regulon (including *ugpB*, *sufA*, *aldB*, *ynhA* and *tal* as well as the aforementioned *sitAB*, *cirA*, *entF*, *fepA*, *iroB*, *entB* targets) was detected, as expression of their corresponding target protein products were found enriched at 24 hpi (FDR $\leq$ 0.01) (Fig 4A and S6 Table). In line with the studies by Liu *et al.* [36] and Li *et al.* [33], bacterial proteome adaptations revealed downregulation of chemotaxis (e.g., CheAMW) and upregulation of His biosynthesis (HisABCDFGHI). In agreement with the increased proliferation rate of cytosolic *Salmonella*, ribosomal proteins (rplBEKMU and rpsCDEJLMRST) and proteins implicated in cell division (MinE, MpI, MukB, MurAEFG, Pal, TolB) were found upregulated at later timepoints of the infection (S6 Table and Fig 5). Since extensive transcriptional reprogramming was shown to accompany the successful colonization of the epithelial cytosol, a niche occupied by a hyper-replicating *Salmonella* sub-population especially at later times post-infection, it is noteworthy that upregulation of cytosol signature genes previously reported to be transcriptionally upregulated in Gram-negative pathogens colonizing the cytosol were also regulated in our proteome study (i.e., *bioA*, *entF*, *fepA*, *fepB*, *iroB*, *iroN* and *sitAB*). Furthermore, Powers *et al.* [59] demonstrated that a *S.* Typhimurium *ΔsitAΔmntH* mutant was compromised for growth in the cytosol of epithelial cells, linking this to the need for acquiring $Mn^{2+}$ when residing in this subcellular location. Altogether, proteome profiling of infecting salmonellae is heavily biased towards its virulence repertoire and provides a proteomic perspective on host-pathogen interactions and (sub-)cellular pathogen localisation over the time course of infection.

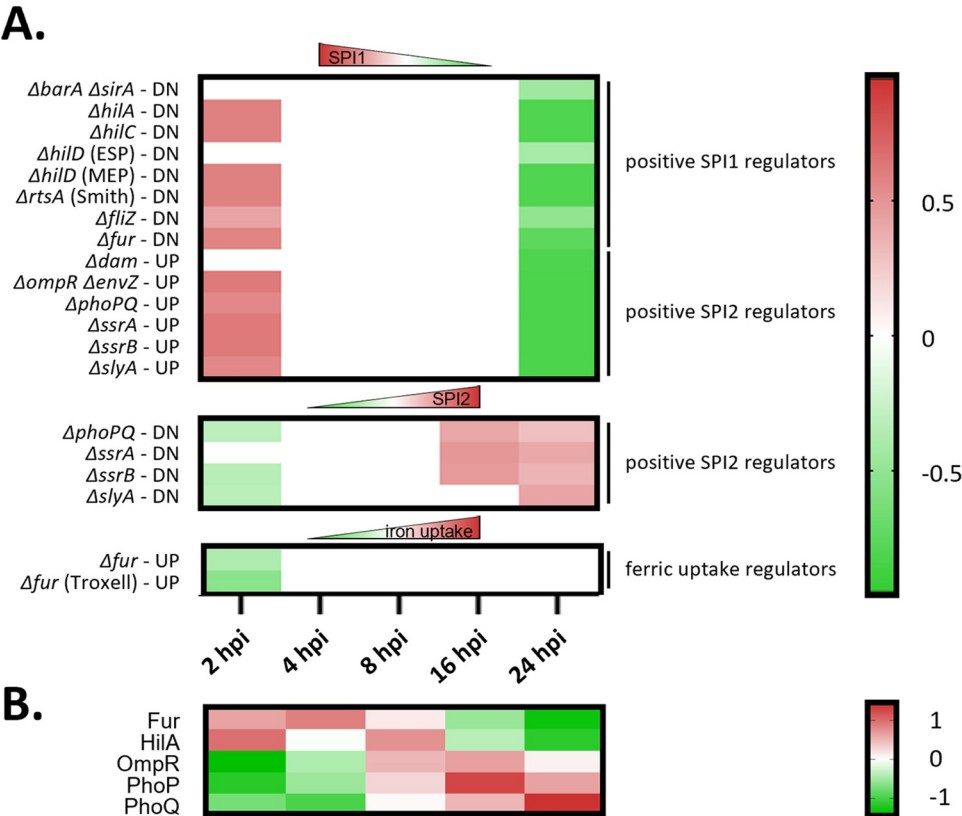

**Fig 4. Heatmaps of significantly regulated (FDR ≤ 0.05) normalized *S*. Typhimurium regulon enrichment scores and expression changes of their identified regulators across all infection conditions (averaged expression values) assayed by data-independent acquisition (DIA). (A)** Using a recently reported compilation of highly curated regulon gene targets identified by means of microarray or NGS-based gene expression studies [59], enrichment of specific regulons was determined using the 1D annotation enrichment algorithm embedded in the Perseus software suite. P-values were corrected for multiple hypotheses testing using the Benjamini and Hochberg false discovery rate. Enriched annotation terms (FDR ≤ 0.05) in the categories are shown as heatmaps and colour coded according to the 1D enrichment scores calculated [49], with red and green colouring indicating enriched and depleted annotations, respectively. ESP (early stationary growth phase), MEP (mid exponential phase), upregulated (UP) and downregulated (DN) genes as reported in [59]. **(B)** Heatmap representation of significantly regulated *S*. Tyhimurium regulators (FDR ≤ 0.01, corresponding normalized averaged z-scores) from the DIA analysis and corresponding to the regulated regulons shown in A. SsrB was also identified, but expression was not significantly regulated.

## Discussion

Given the generally acknowledged multifactorial complexity of host-pathogen interactions, the data variability in infection studies reported, and the high temporal aspect of the interactions, multiplex omics-based analyses deliver a comprehensive understanding of the complexity of such systems and the interplay between pathogen and host. To understand bacterial adaptation to host encounters, to date, the vast majority of *Salmonella* centred proteomics research relied on axenic grown *Salmonella* exposed to various (infection-relevant) growth conditions [61]. Only more recently, extensive adaptation of *S*. Typhimurium to infected host epithelial cells was shown by proteomics efforts making use of enriched *Salmonella* isolated from infected epithelial host cells [33, 36, 60]. The aforementioned strategies however requiring a rather lengthy and potentially biased enrichment procedure for the isolation of intracellular bacteria. Here, we report for the first time a dual proteome profile of *Salmonella*-infected epithelial host cells in function of time, both in DIA and DDA modes, overall providing a proteomic

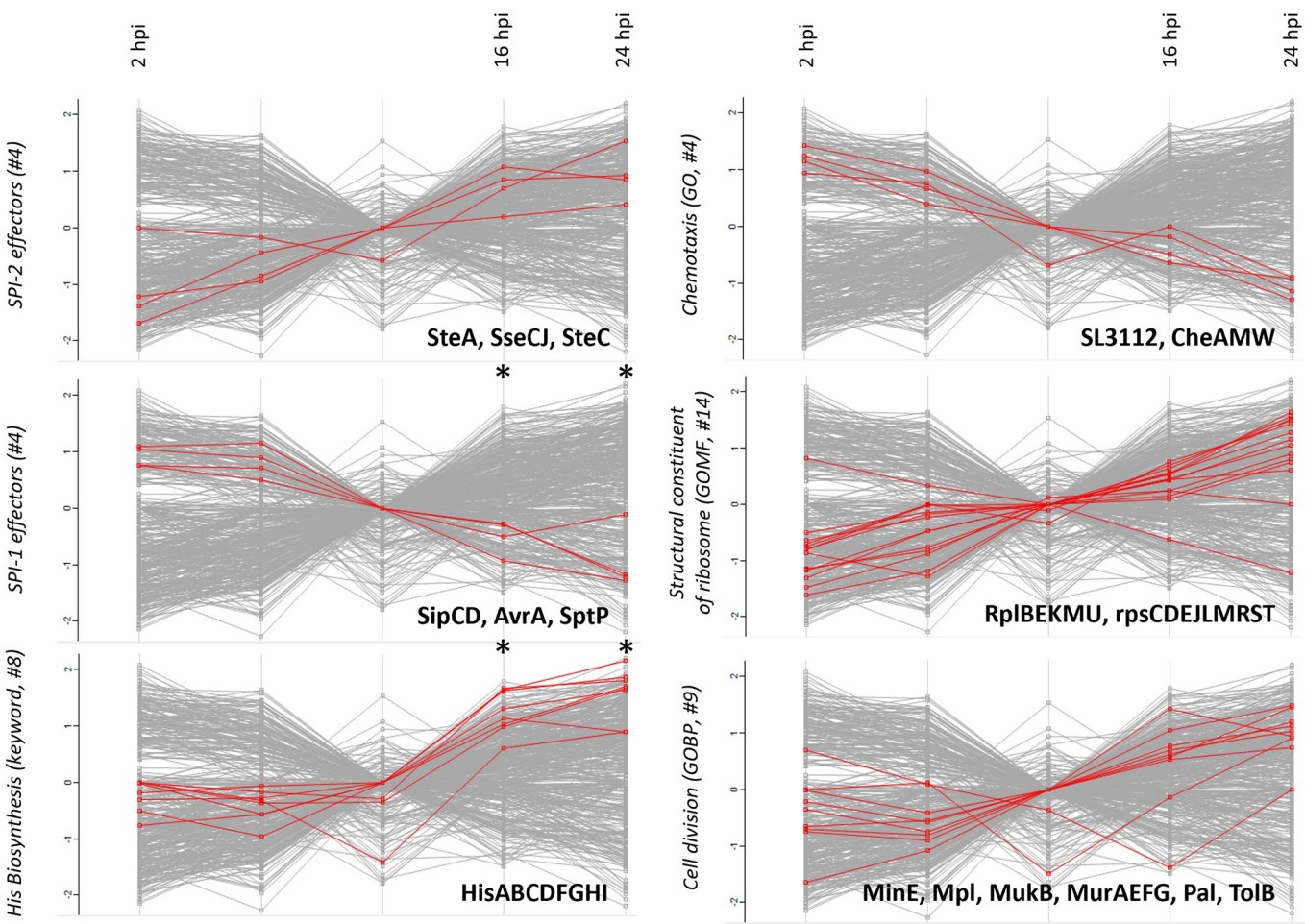

**Fig 5. Protein profile plots of significantly regulated (FDR ≤ 0.05) normalized *S*. Typhimurium annotation enrichment scores across all 5 infection conditions (averaged expression values) assayed by data-independent acquisition (DIA).** Term enrichment was determined using the 1D annotation enrichment algorithm embedded in the Perseus software suite and p-values were corrected for multiple hypotheses testing using the Benjamini and Hochberg false discovery rate. Only corrected p-values ≤ 0.05 were considered. Selected GO terms and keyword annotations for with significantly altered protein abundances were observed (i.e., SPI-1 (#4); downregulation at 16 and 24 hpi, histidine biosynthesis (keyword, #8); upregulation at 16 and 24 hpi) are shown for the 5 infection setups analysed. A black asterisk above the profile plots means significant upregulation (by 1D annotation enrichment of the corresponding proteins in the annotation group. Besides, some other categories (SPI-2 (#4), chemotaxis (GO, #4), structural constituent of the ribosome (GOMF, #14) and cell division (GOBP, #9) of which the corresponding protein members were all found to be significantly regulated based on t-testing are shown as profile plots.

perspective on the bacterial pathogen encounter with its epithelial host and permitting the simultaneous study of host and pathogen proteomes.

Noteworthy, while DIA outperformed DDA both in terms of identifications and quantifications, especially for the bacterial pathogen, for DDA data, correlation of quantifications was shown to be higher than for DIA data. This finding is mainly attributed to the increased DIA-based quantification of low abundant peptides and due to the higher variability of quantification data for low abundance species given the interference with random noise [31].

This observation is also apparent from the generally less discrete (differential) patterns of *Salmonella* protein expression as compared to the distinctive human expression clusters. The former finding is also likely (in part) due to the biological variability of proteome sampling at different timepoints post-infection.

Bacterial proteome sampling in dual proteome profiling nevertheless remains less comprehensive as compared to previously reported protein identifications making use of pure

*Salmonella* cultures or of *Salmonella* isolated from infected epithelial host cells. For example, when considering axenic *S.* Typhimurium cultures and triplicate shotgun proteome analysis, typically a 2-fold increase in the number of *S.* Typhimurium proteins identified and quantified [62] (around ~1,500 to 1,600) are found using similar MS instrumentation in DDA mode [30]. Noteworthy, while a significant higher number of *Salmonella* proteins were reported (~1,800) by Li et al making use of enriched *Salmonella* [33], it is important to note that samples were extensively fractionated by means of 1D SDS-PAGE separation and slicing of the gel into 36 gel pieces per biological replicate sample, while in our study unfractionated samples were analysed using a single LC-MS run per replicate sample. As such, offline sample fractionation in combination with DIA can provide a useful asset to improve DIA identification rates, especially in the case of complex proteome mixes with components of low abundance [31]. Alternatively, as demonstrated in the EncyclopeDIA workflow, making use of parallel DIA runs with narrow *m/z* windows in consecutive *m/z* ranges [48], a significant increase in the identification of *Salmonella* proteins can be expected in case of dual proteome profiling. Altogether, this indicates that sensitivity does not represent an inherent major shortcoming of dual proteome profiling.

Further, the high quantification sensitivity obtained by DIA permitted the reliable capture of over 100 regulated bacterial proteins over the time course of infection, generally exceeding the numbers of regulated bacterial genes reported in other studies, and, importantly, also including the identification of downregulated *Salmonella* proteins upon infection.

In line with the relative comparisons of *Salmonella* protein expression at various timepoints post-infection previously reported [36] and by comparing protein expression regulation across infection conditions only, initial bacterial adaptations to the host might be missed. However, upregulation of e.g., phosphate utilization, previously shown to be highly induced in the *Salmonella* proteome upon medium switching occurring prior to bacterial internalization [60], was not observed in our study. This indicates that likely phosphate shortage is not experienced within the host itself but rather dependent on bacterial cultivation steps preceding the actual infection, necessitating the need for detailed experimental descriptions and uniformity when comparing omics datasets, hence the value of dual omics strategies.

On a related note, upon systematically comparing our dual proteomics data with previously reported transcriptome changes making use of dual RNA-sequencing and FACS-enriched *S.* Typhimurium infected HeLa cells [41], for the about 5,700 matching identifications (DIA data), transcriptome versus proteome changes were found to poorly correlate, especially at later timepoints post-infection (Pearson correlation below 0.2 at 24 hpi). Annotation enrichment analysis even pointed to oppositely regulated gene expressions when comparing protein versus transcript levels. In particular, while proteome data indicates downregulation of the GO term 'actin cytoskeleton' at 24 hpi, upregulation was observed at the transcript level. Opposingly at 16 and 24 hpi, upregulated ribosome biogenesis and mitochondrial gene expression was observed at the proteome level, while transcript levels were down (S7 Fig). Overall, this indicates that besides the lag phase between transcription and translation to be considered, translational and/or post-translational control significantly impacts bacterial gene expression especially in the context of (prolonged) infections, further emphasizing the need to study gene expression changes at the proteome level.

Further noteworthy in this context is that *S.* Typhimurium T3E host target proteins were found enriched in the category of regulated proteins as determined by a chi-square test of independency (p < 0.01; p-value 0.002203), indicating that bacterial effector functioning greatly shapes the proteome of the host. As a representative example, it has been shown that *Salmonella* remodels the host actin cytoskeleton to force its entry into non-phagocytic cells. These cytoskeletal rearrangements are mediated by T3SS-1 effectors such as SopB, SopE and

SopE2, with indirect actions that are essential for efficient invasion [63]. Several other effectors were shown to directly interact with actin(-binding) proteins including effector SspH2 that has established interactions with profilin (PFN)-1 and -2 inside the host [57, 64, 65]. In our DIA data, PFN2 showed to be ~3-fold upregulated at 2 hpi. With its predicted inhibitory effect on the formation of membrane ruffles, PFN2 upregulation at 2 hpi supports the predicted down-regulation of "formation of cellular protrusions" when performing Ingenuity Pathway Analysis (IPA) (S8A Fig).

To restore the host cell cytoskeleton, *Salmonella* effector SptP uses its GTPase-activating protein (GAP) activity to inactivate RAC1 [66], thereby reversing the Rho GTPase activating effects of SopE [67]. This restoration of the host cytoskeleton is also evident from the predicted activation of the "formation of actin filaments" at 2 hpi and the regulation of contributing genes (S8A and S8B Fig).

Further, IPA pathway analysis substantiated the observation that apoptosis of epithelial cells infected with *Salmonella* is delayed until 8 hpi (S8A Fig) [68], an observation in line with our host cell viability data. This delay in apoptosis is thought to be attributed to sustained AKT signalling that is accredited to the multifaceted effector SopB, a lipid phosphotransferase [69].

Further, in order for *Salmonella* to manipulate the host endocytic pathway, several ras-asso-ciated binding (RAB) proteins have been reported to be directly targeted by effectors (e.g., GtgE, SopD and SopD2 [70–72]. Based on our IPA analysis, autophagy has a positive activa-tion z-score at 8 hpi (S8A Fig) with a predicted contribution of the RAB1A, RAB7A and RAB9A, all found at higher abundancies at this time point in our DIA data.

Next to demonstrating the fluctuations among functions affected during *Salmonella* infec-tion, IPA revealed 60 putative upstream regulators to be either activated or inhibited over the course of infection (S8C Fig). Among others, immune system signalling regulators such as toll-like receptor signalling proteins, e.g. STAT1, IL1B, IFNA2 and TLR3 and MAPK3, were shown to be activated at 16–24 hpi.

Viewing the complex response to bacterial infection observed over time, and with some of the expression differences linked to intracellular host localization of the bacteria and shown to promote intracellular *Salmonella* growth [36], studies on the causal links between host/patho-gen expression interdependencies remain an interesting area of future research. For this, strain-specific responses of the host to *Salmonella* infection (considering an e.g., SPI-2 mutant) could be interrogated to uncover unique profiles of pathogen/host interdependencies over the time course of the infection.

Moreover, with the observed variability of *Salmonella* subpopulations in diverse host cells [23, 24], future DIA-based dual proteome profiling infection studies performed at the sub-population or even single-cell level may further advance our understanding of host-pathogen interactions at an unprecedented resolution.

## Materials and methods

### Bacterial strain and growth cultivation conditions

The *Salmonella enterica* serovar Typhimurium (*S*. Typhimurium) wild-type (WT) strain SL1344 (Hoiseth and Stocker, 1981) (Genotype: hisG46, Phenotype: His(-); biotype 26i) was obtained from the *Salmonella* Genetic Stock Center (*Salmonella* Genetic Stock Center (SGSC), Calgary, Canada; cat n˚ 438 [73]). The GFP-expressing *Salmonella* SL1344 strain JVS-3858 with P*tet*::*gfp* integrated into the *put* locus of the chromosome [45] was a gift from Prof. Jörg Vogel and Prof. Alexander Westermann. Bacterial growth was performed in Luria Beltrami (LB)-Miller broth (10 g/L Bacto tryptone, 5 g/L Bacto yeast extract, 10 g/L NaCl). For bacterial cultivation, single colonies were picked from LB plates, inoculated in 3 ml liquid Lennox (L)

growth medium (L-broth) in round-bottom falcon tubes (Corning, cat n˚ 352059) and grown for ~18 h at 37˚C with agitation (180 rpm). Subsequently, the cultures (~$OD_{600}$ 2.0) were diluted 1:50 in T25 flasks with ventilated cap in 8 ml L-medium and grown at 37˚C and 180 rpm to early stationary phase ($OD_{600}$ 3.0) (corresponding to ~$2 \times 10^9$ bacteria per ml and ~3.5 h of culture after 1/50 dilution of an overnight culture).

## Cell culture

Human epithelial HeLa cells (epithelial cervix adenocarcinoma cell line, American Type Culture Collection, Manassas, VA, USA; ATCC *CCL2*) were cultured in GlutaMAX containing Dulbecco's Modified Eagle Medium (DMEM) (Gibco, cat n˚ 31966–047) supplemented with 10% foetal bovine serum (FBS) (Gibco, cat n˚ 10270–106) and 50 units/ml penicillin and 50 μg/ml streptomycin (Gibco; cat n˚ 5070–063) (0.22 μm filtered in case of real-time imaging in 96-well plates (see below methods)). Cells were cultured at 37˚C in a 5% $CO_2$ gas-equilibrated humidified incubator and passaged every 3–4 days.

For infection, HeLa cells were seeded in a T175 (Greiner Bio One, cat n˚ 660160) at a density of $10^7$ cells in 19 ml of medium for dual-proteome profiling, or at $1.5 \times 10^4$ cells per well in 96-well plates (Tecan 96 Flat Black, cat n˚ 30122306) in 150 μl of medium for cell-free cell-counting and fluorescence detection. The culture medium was supplemented with 10% heat-inactivated FBS (heat inactivation was done for 30 min at 56˚C) and without antibiotics. In both set-ups, HeLa cells were seeded one day prior to infection thereby reaching a final confluence of ~85–90%, corresponding to approx. $1.25 \times 10^7$ HeLa cells per T175 cell culture flask and $1,875 \times 10^4$ HeLa cells per well in 96-well plates at the day of infection.

## Bacterial infection of cultured HeLa cells

For bacterial invasion, the bacteria were harvested for infection at early stationary phase (when studying *Salmonella* spp. Invasion of epithelial cell lines, the early stationary phase was shown to be the phase during which *Salmonella* pathogenesis island-1 (SPI-1) is highly induced, and therefore a suitable growth phase for infection [74]) after reaching an optical density at 600 nm of 3.0. Bacterial cells were collected by centrifugation ($6,000 \times g$, 10 min) at room temperature, washed once with pre-heated (37˚C) Dulbecco's Phosphate Buffered Saline (DPBS) (Gibco; cat n˚ 14190–144), re-spun and resuspended in pre-heated DMEM without serum.

*Salmonella* infection was allowed to proceed in DMEM at a multiplicity of infection (MOI) of 100 (i.e., per T175 cell culture flask of $1,25 \times 10^7$ HeLa cells, $1,25 \times 10^9$ bacterial cells (or 1.25 ml of a $10^9$ bacteria bacterial cell suspension)), or in case of 96-wells, 50 μl containing $1,875 \times 10^5$ (MOI 10), $1,875 \times 10^6$ (MOI 100) or $3,75 \times 10^6$ bacterial cells (MOI 200) were added for 30 min at 37˚C in a 5% $CO_2$ gas-equilibrated humidified incubator. After infection, the medium was removed from the culture flasks or the 96-well plates, the monolayers washed with DPBS pre-heated at 37˚C and respectively, 20 ml or 0,2 ml of pre-warmed DMEM supplemented with heat-inactivated FBS and 100 μg/ml gentamicin (Sigma-Aldrich, G1264-1G) was added to kill extracellular bacteria after which the infection was allowed to proceed for an additional 2 h. When the infection was allowed to proceed for longer times, after 2 h, cells were washed and DMEM supplemented with 10 μg/ml gentamicin was added. For dual proteome profiling, 4 replicate samples corresponding to 2, 4, 8, 16 and 24 hours post-infection (hpi) were collected, besides a HeLa control setup. After a DPBS wash (37˚C) of the cell monolayers, cells were harvested by trypsinization (Trypsin-EDTA; Gibco, cat n˚ 25300054), collected by centrifugation ($1000 \times g$, 5 min) at 4˚C and the cell pellets washed 2 times with DPBS, flash frozen in liquid nitrogen and stored at -80˚C until further processing.

## Real-time assessment of host cell count and death, besides monitoring of *Salmonella* infection

For real-time measurement of label-free host cell count (indicative of the overall viability of infected cell cultures) and cell death, besides monitoring of *Salmonella*, 4 replicates samples infected with either WT SL1344, a constitutively eGFP-expressing derivative strain of SL1344 (i.e., JVS-3858 [45]), and a non-invasive Δ*prgH* mutant at different MOIs (MOI 10,100 and 200) were analysed over a 24 hours' time course of infection. Replicates to which CellTox Green Dye was added (Promega; Cat. n° G873A) at 4x concentration according to the manufacturer's instructions were analysed in parallel. To calibrate fluorescence read-out, a lysis control was prepared by adding 8 µl lysis buffer (Promega; Lysis Solution; G182B) (1/25) to control cell cultures. Plates were sealed with sterile, gas-permeable, optical adhesive sealing film for microplates and incubated in a small humidity cassette of a Spark Cyto 400 multi-well optical plate reader, controlled by SparkControl V3.2 software (Tecan, Mechelen, Belgium). Cultures were monitored for 24 hours and the image acquisition within the Spark Cyto performed using a user-defined application in SparkControl. A bright field center image was captured every 30 min for each well using a 10x objective (center grid, user-defined area) and fluorescence was measured every ~30 min (sensitivity 70%, object length and width ranging between 8–30 µm, analysis type: segmentation), and the count and confluence algorithms used to calculate the cell count and cell-covered area, respectively. The sensitivity of these algorithms were further refined post-acquisition using the Spark Cyto's image analysis software; Image Analyzer. GFP and CellTox Green fluorescence was measured using bottom reading and an excitation and emission wavelength of 485 nm (width 20 nm) and 535 nm (width 20 nm), respectively.

## *prgH* deletion mutant generation by λ Red recombineering

A *S.* Typhimurium SL1344 *prgH* mutant (Δ*prgH*) was constructed by λ Red recombineering [75]. Briefly, ~2.4 x $10^9$ of λ red-induced (0.2% L–(+)–arabinose w/v) electrocompetent and pKD46 (accession number #7669, Coli Genetic Stock Center (CGSS), Yale University, USA) transformed *S.* Typhimurium WT cells (culture grown till $OD_{600}$ 0.6) were electroporated with a linear PCR-editing substrate designed to replace *prgH* with the kanamycin-resistance cassette from pKD4 (accession number #7632, CGSS, Yale, USA). PCR was used to amplify the antibiotic-resistance cassette with 5' and 3' 50 bp homology arms complementary to the flanking regions of *prgH* using primer sequences caatggggatgatggttcttttaatatgtgttgagacgcattatacagaat*gtgtaggctggagctgcttc* and acgccgccagtagcgccggatcggagggttttgctgctaatttatccagctatgaatatcctccttagttcctatt). Immediately following electroporation, bacterial cells were recovered in pre-warmed (28°C) Super Optimal broth with Catabolite repression (SOC, consisting of 2% tryptone, 0.5% yeast extract, 10 mM NaCl, 2.5 mM KCl, 5 mM $MgCl_2$, 10 mM $MgSO_4$ and 20 mM glucose) media, incubated at 180 rpm for 1 h at 28°C and subsequently at 37°C for 3 h. Mutant colonies were selected after plating the cell suspension and overnight incubation (upside down) on LB agar plates supplemented with 25 µg/ml kanamycin at 37°C. Allelic replacement of *prgH* was confirmed by colony PCR using primer sequences ggatgatggttcttttaatatgt and ggcaagaaagccatccagtttactttgca, and acgccgccagtagcgccggatcg and gacgagttcttctgagcgggact, and the resulting PCR products sequence verified by Sanger sequencing (Applied Biosystems 3730xl, Thermo, VIB Genomic Service Facility, University of Antwerp, Belgium).

## Proteome extractions and sample mixes

Cell pellets of 4 replicate samples and 6 setups (control HeLa + 2, 4, 8, 16 and 24 hpi samples) were resuspended in guanidinium-hydrochloride (Gu.HCl) lysis buffer at 20 x $10^6$ cells/ml

(4 M Gu.HCl, 50 mm ammonium bicarbonate (pH 7.9)) and lysed by three rounds of freeze-thaw lysis in liquid nitrogen. The lysates were sonicated (Branson probe sonifier output 40, 50% duty cycle, 3×30 s, 1 s pulses) followed by centrifugation (16,100 x *g*, 10 min at 4˚C) to remove cellular debris. The protein concentration of the supernatant was determined by Bradford measurement [76] according to the manufacturer's instructions (Bio-Rad, cat n˚ 5000006).

An aliquot adjusted with lysis buffer to 2 mg/ml, and equivalent of 400 μg of total protein (200 μl) was transferred to a clean Eppendorf tube, twice diluted with HPLC-grade water, and precipitated with 4 volumes of acetone (at-20˚C) overnight. The precipitated protein material was recovered by centrifugation for 15 min at 16,000 x *g* at 4˚C, pellets washed twice with cold 80% acetone, and air dried upside down for ~10 min at room temperature or until no residual acetone odour remained. Protein pellets were resuspended in 200 μl TFE (2,2,2-trifluoroethanol) digestion buffer (10% TFE, 100 mM ammonium bicarbonate pH 7.9) with sonication (Branson probe output 20; 1 s pulses) until a homogenous suspension was obtained. All samples were digested overnight at 37˚C using MS-grade trypsin/Lys-C Mix (Promega, Madison, WI) (enzyme/substrate of 1:50 w/w) while mixing (550 rpm). Samples were acidified with TFA to a final concentration of 0.5%, cleared from insoluble particulates by centrifugation for 10 min at 16,000 x *g* at 4˚C and the supernatant transferred to new tubes. Methionine oxidation was performed by addition of hydrogen peroxide to a final concentration of 0.5% for 30 min at 30˚C. Solid phase extraction of peptides was performed using C18 reversed phase sorbent containing 100 μl pipette tips Bond Elut OMIX 100 μl C18 tips (Agilent, Santa Clara, CA, USA, cat n˚ A57003100K) according to the manufacturer's instructions. The pipette tip was conditioned by aspirating the maximum pipette tip volume of water:acetonitrile, 50:50 (v/v) and the solvent discarded. After equilibration of the tip by washing three times with the maximum pipette tip volume in 0.1% TFA in water, 100 μl of the acidified samples (original input of ~200 μg) were dispensed and aspirated for 10 cycles for maximum binding efficiency. The tip was washed three times with the maximum pipette tip volume of 0.1% TFA in water:acetonitrile, 98:2 (v/v) and the bound peptides eluted in LC-MS/MS vials with the maximum pipette tip volume of 0.1% TFA in water:acetonitrile, 30:70 (v/v). The samples were vacuum-dried in a SpeedVac concentrator and re-dissolved in 50 μl of 2 mM tris(2-carboxyethyl)phosphine (TCEP) in 2% acetonitrile spiked with an indexed Retention Time or iRT peptide mix (i.e., a mixture of eleven non-naturally occurring synthetic peptides added according to manufacturer's instructions (Biognosys)), to enable retention time (RT) prediction (see below). Samples were stored at -20˚C until LC-MS/MS analysis.

## LC-MS/MS data acquisition

For mass spectrometry analyses, 10 μl was injected from each sample for LC-MS/MS analysis on an Ultimate 3000 RSLCnano system in-line connected to a Q Exactive HF Hybrid Quadrupole-Orbitrap BioPharma mass spectrometer (Thermo). Trapping was performed at 10 μL/min for 4 min in loading solvent A (0.1% TFA in water/I (98:2, v/v) on a 20 mm trapping column (made in-house, 100 μm internal diameter (I.D.), 5 μm beads, C18 Reprosil-HD, Dr. Maisch, Germany). After flushing from the trapping column, the peptides were loaded and separated on an analytical 200 cm μPAC column with C18-endcapped functionality (Pharma-Fluidics, Belgium) kept at a constant temperature of 50˚C. Peptides were eluted by a non-linear gradient reaching 9% MS solvent B (0.1% FA in water/acetonitrile (2:8, v/v)) in 15 min, 33% MS solvent B in 105 min, 55% MS solvent B in 125 min and 99% MS solvent B in 135 min at a constant flow rate of 300 nL/min, followed by a 5-minutes wash at 99% MS solvent B and re-equilibration with MS solvent A (0.1% FA in water). For both analyses (data-dependent

acquisition (DDA) and data-independent acquisition (DIA), a pneu-Nimbus dual column ionization source was used (Phoenix S&T), at a spray voltage of 2.6 kV and a capillary temperature of 275°C. For the first analysis (DDA mode), the mass spectrometer automatically switched between MS and MS2 acquisition for the 16 most abundant ion peaks per MS spectrum. Full-scan MS spectra (375–1500 *m/z*) were acquired at a precursor resolution of 60,000 at 200 *m/z* in the Orbitrap analyser after accumulation to a target value of 3,000,000. The 16 most intense ions above a threshold value of 13,000 were isolated for higher-energy collisional dissociation (HCD) fragmentation at a normalized collision energy of 28% after filling the trap at a target value of 100,000 for maximum 80 ms injection time using a dynamic exclusion of 12 s. MS2 spectra (200–2,000 *m/z*) were acquired at a resolution of 15,000 at 200 *m/z* in the Orbitrap analyser. Another equivalent 10 μL aliquot from each sample was analysed using the same mass spectrometer now operated in DIA mode. Nano LC conditions and gradients were the same as used for DDA. Full-scan MS spectra ranging from 375–1,500 *m/z* with a target value of 5E6 were followed by 30 quadrupole isolations with a precursor isolation width of 10 m/z for HCD fragmentation at a normalized collision energy of 30% after filling the trap at a target value of 3E6 for maximum injection time of 45 ms. MS2 spectra were acquired at a resolution of 15,000 at 200 m/z in the Orbitrap analyser without multiplexing. The isolation intervals ranged from 400–900 *m/z* with an overlap of *m/z*.

## Data-dependent acquisition (DDA) data processing

For DDA-based peptide identification and quantification, Raw data files were searched by MaxQuant [47] (version 1.6.9.0). The protein search database comprises the UniProtKB *Salmonella* reference proteome concatenated to the human UniProtKB reference proteome (UP000008962 and UP000005640, together 74,449 proteins), as well as MaxQuant built-in contaminants. In addition, contaminant proteins present in the contaminants.fasta file that comes with MaxQuant and the 11 iRT peptide sequences (Biognosys-11) were added to the search database. Methionine oxidation was set as fixed modification and N-terminal acetylation was set as variable modification. To augment protein/peptide quantification, we enabled built-in matching-between-run and label-free quantitation (LFQ) algorithms using default settings in MaxQuant. We used the enzymatic rule of trypsin/P with a maximum of 2 missed cleavages. The main search peptide tolerance was set to 4.5 ppm and the ion trap MS/MS match tolerance was set to 0.5 Da. Default PSM, peptide and protein level FDR thresholds of 1% were used, with an additional minimal Andromeda score of 40 for modified peptides. Protein FDR was set at 1% and estimated by using the reversed search sequences. The maximal number of modifications per peptide was set to 5. For protein quantification in the proteinGroups.txt file, razor peptides were considered and further allowed all modifications.

## Data-independent acquisition (DIA) data processing

**DDA-based spectral library construction.** The 'msms.txt' files outputted by MaxQuant DDA searches were used as input for creation of redundant BLIB spectral libraries using BiblioSpec (version 2.1) [77]. Redundant spectra were subsequently filtered using the 'BlibFilter' function, requiring entries to have at least 5 peaks ('-n 5'). Peptide sequences uniquely identified in only one of the DDA searches were appended to the spectral library to extend the library and to augment the detection capacity of *Salmonella* peptides. We transformed the peptide RTs present in the BLIB library to iRT using the spiked-in iRT peptides (Biognosys-11). To this end, empirical RTs of the top-scoring iRT peptide identifications (lowest posterior error probability, 'msms.txt') in the DDA samples, were used to fit a linear trendline and scale RTs. For the artificial mixtures and *Salmonella* pre-fractionation analyses [31], the

corresponding trendlines were iRT = 1.220 RT– 74.566 and iRT = 1.189 RT– 75.039. The updated BLIB files were then converted to DLIB format by EncyclopeDIA, using the combined human-*Salmonella* UniProtKB proteome FASTA as background [48].

**Library-free searching of proteome FASTA by PECAN/Walnut.** DIA raw data files were converted to mzML by MSConvert using vendor peakPicking. Pre-processed DIA samples were searched against a compilation of the *Salmonella* UniProtKB proteome (UP000008962, 4,657 proteins) and human Swiss-Prot proteome (UP000005640, 20,367 proteins) using the EncylopeDIA built-in PECAN algorithm [48, 78]. We opted to solely search the human Swiss-Prot protein database, which resulted in a ~3-fold reduction of the protein database search space (25,024 versus 74,449 proteins when combining *Salmonella* [UP000008962] and human [UP000005640] UniProtKB references proteomes), in order to minimize the theoretical search space. This was desired to limit the size of a predicted spectral library for all possible tryptic peptides and overall runtime and memory usage. Since 36,494 out of 36,668 (i.e., 99.53%) of all human peptides identified by MaxQuant matched a Swiss-Prot protein entry, no drastic loss in identifications is anticipated. Default settings were used, except for methionine oxidation (to methionine-sulfoxide) being set as fixed modification, and considering a maximum length of 25 amino acids and HCD as fragmentation type.

**Construction of an MS²PIP-based spectral library.** MS2 spectra were predicted by MS²PIP (version 20190312) [79] for tryptic peptides derived from an *in silico* digest of the *Salmonella* UniProtKB proteome and human Swiss-Prot proteome (trypsin/P, peptide length 7–25 AA, mass 500–5,000 Da, one missed cleavage, N-terminal initiator methionine removal considered) in case of 2+ and or 3+ peptide precursor fit within 400 to 900 *m/z* (scanned range DIA). This yields a total of 1,586,777 predicted MS2 spectra for 1,151,386 peptides solely matching human proteins, 197,782 spectra for 144,156 peptides matching *Salmonella* proteins, and 117 spectra for 110 peptides matching both species. We set methionine oxidation (to methionine-sulfoxide) as a fixed modification for MS2 prediction by MS²PIP. Predicted spectra were supplemented with DeepLC predicted RTs using a model trained on RTs of 35,206 non-redundant peptides identified in DIA PECAN searches (peptide Q-value < 0.01) (as described in the above section).

**Hybrid library construction.** The chromatogram libraries (ELIB) generated by EncyclopeDIA after PECAN and MS2PIP-library searching were combined into a single redundant library. After conversion to BLIB format, and adding the ':redundant:' tag in library info (sqlite3), the 'BlibFilter' function was used to create a non-redundant DIA spectral library as described above. This DIA spectral library was further extended with spectra from the extended DDA spectral library (see above) corresponding to peptides not yet contained within the DIA spectral library. The resulting hybrid DDA/DIA spectral library (i.e., joining DDA- and DIA-based identifications) was then converted to DLIB format as described above.

**EncyclopeDIA spectral library searching and peptide quantification.** The resulting mzML files were searched against the DDA, MS²PIP, or hybrid spectral DLIB libraries using EncylopeDIA software (version 0.90) [48] with default settings. Sample-specific Percolator output files and chromatogram libraries were stored. Per setup, a combined chromatogram library was created consisting of the three replicates. This performs a Percolator re-running of the combined results and provides peptide and protein quantifications at a 1% peptide and protein Q-value, respectively. For quantification, the number of minimum required and quantifiable ions were set at 5 with aligning between samples enabled.

MS2PIP, Elude and EncyclopeDIA are open source, licensed under the Apache-2.0 License, and are hosted on https://github.com/compomics/ms2pip_c, https://github.com/percolator/percolator and https://bitbucket.org/searleb/encyclopedia/wiki/Home.

## Software packages for statistics and data visualization

For basic data handling, normalization, statistics (if not stated otherwise) and annotation enrichment analysis, we used the freely available open-source bioinformatics platform Perseus (http://141.61.102.17/perseus_doku/doku.php?id=start) (v1.6.5.0). Perseus was used to analyse data from principal component analysis, non-supervised hierarchical clustering, and scatter plots. Furthermore, the 1D and 2D annotation algorithms and Fisher's exact tests implemented in Perseus (Cox & Mann, 2012) were used for annotation enrichment analysis. Heat maps, PCA plots and bar charts were generated using GraphPad Prism software (www. graphpad.com).

## IPA Pathway analysis

For the different time points post-infection, significantly regulated human proteins (t-test, p-value $\leq$ 0.01; described above) were characterized by core analysis in IPA software (version 68752261, QIAGEN Inc., https://www.qiagenbioinformatics.com/products/ingenuity-pathway-analysis) using the t-test differences as directional expression values. Upstream regulators were determined using a p-value of overlap (cutoff set at 0.01) and z-scores calculated using the algorithm implemented in IPA [80].

## Supporting information

**S1 Fig. Characterization of epithelial *Salmonella* infection model.** HeLa cells (1,875 x $10^4$ HeLa cells) were non-infected or infected with *S.* Typhimurium (WT or Δ*prgH* SL1344) at MOIs 10 (blue), 100 (red) or 200 (green) in quadruplicate and cell count and cytotoxicity (or GFP fluorescence) measured in a multi-well optical plate reader in real-time. Label-free cell counting (10x objective) of HeLa cell cultures infected with WT *Salmonella* (**A**) or non-invasive Δ*prgH* mutant *Salmonella* (**B**). Monitoring cell death in real-time using the CellTox Green Cytotoxicity Assay (**C-D**). Using a constitutively eGFP-expressing strain of SL1344 [45] (WT eGFP) green objects were counted and their average size (μm$^2$) determined (**F**). Due to technical measurement errors, the 17.5 and 18 hpi recordings were omitted.
(TIF)

**S2 Fig. Variability of DDA and DIA replicate samples represented in PCA plots.** Individual DDA (**A-C**) and DIA (**B-D**) PCA plots for *Salmonella* (**C-D**) as well as HeLa (**A-B**) over the time-course of infection. Green, cyan, blue, orange, purple and black circles represent control (Ctrl), 2, 4, 8, 16 and 24 hpi samples, respectively.
(TIF)

**S3 Fig. Correlations (Pearson) between all setups and replicate samples analysed.** Ctrl HeLa, 2 hpi, 4 hpi, 8 hpi, 16 hpi, and 24 hpi.
(TIF)

**S4 Fig. Average mean correlations (Pearson) between setups analysed.** Ctrl HeLa, 2 hpi, 4 hpi, 8 hpi, 16 hpi, and 24 hpi.
(TIF)

**S5 Fig. Profile plots of regulated *S.* Typhimurium proteins over the time-course of infection.** Significant regulation (FDR $\leq$ 0.01) was determined across the 5 timepoints post-infection using multiple t-testing and corresponding normalized averaged z-scores of *S.* Typhimurium DDA (left panels) and DIA data (right panels) plotted, with corresponding members of the two main clusters (i.e., up- and downregulated cluster) shown and its

corresponding number of regulated *S.* Typhimurium protein groups indicated.
(TIF)

**S6 Fig. Profile plots and heat map representation of regulated human proteins over the time-course of infection.** Regulation was determined across the 5 timepoints post-infection and control cells using t-testing for normalized DIA data. The intensities of significantly (FDR ≤ 0.01) regulated human proteins (**#1,742**) from the DIA analysis are shown. The averaged expression values of corresponding members of the 7 main clusters after ANOVA are shown as profile plots **(A)** and as a heat map **(B)**, with the corresponding number of regulated protein groups indicated per cluster (see also **S5 Table**). In panel **B**, green indicates low intensities while red indicates high intensities.
(TIF)

**S7 Fig. 2D annotation enrichment of human proteome and transcriptome expression changes.** The 2D-annotation enrichment scores for representative GO biological processes (GOBP), GO cellular component (GOCC), GO molecular function (GOMF) and keywords were plotted for the proteome (DIA data) and transcriptome expression changes (GEO series accession number project GSE60144, [41]) at 2, 4, 16 and 24 hpi (no 2D enriched terms were observed at 8 hpi). The fold change of log2 transformed means of corresponding LFQ and RPKM values were plotted and only corrected *p* values ≤0.01 were considered.
(TIF)

**S8 Fig. Selection of functions and regulators predicted to be activated or inhibited using Ingenuity Pathway Analysis (IPA). (A)** Heatmap visualization (IPA activation z-score) of selected IPA functions regulated over the course of the infection. **(B)** Overview of regulated genes (#85) (log FC) contributing to the IPA function "formation of actin filaments" (see also **S5 Table**). **(C)** Overview of activated and inhibited upstream regulators. Heatmap representation of 60 predicted proteinaceous (activated or inhibited for at least one of the timepoints post-infection analysed) upstream regulators (p-value ≤ 0.01), colour coded according to the IPA activation z-score.
(TIF)

**S1 Table. List of 6,696 proteins identified in the dual proteome shotgun samples of *S.* Typhimurium infected human HeLa cells by means of data-dependent acquisition (DDA).** *S.* Typhimurium infected human HeLa cells were analysed to monitor quantitative differences in steady-state protein expression levels at 2, 4, 8, 16 and 24 hpi in biological quadruplicates. UniProt database primary accession number, UniProt entry (name), gene name, corresponding protein description and sequence, molecular protein weight, number of (razor/unique) peptides identified, (razor/unique) sequence coverage, score, log2-transformed LFQ protein intensities, GO annotation and keywords, MS/MS counts and sequence length(s) are given. Ranking is done alphabetically according to protein names.
(XLSX)

**S2 Table. List of 6,734 proteins identified in the dual proteome shotgun samples of *S.* Typhimurium infected human HeLa cells by means of data-independent acquisition (DIA).** *S.* Typhimurium infected human HeLa cells were analysed to monitor quantitative differences in steady-state protein expression levels at 2, 4, 8, 16 and 24 hpi in biological quadruplicates. UniProt database primary accession number, UniProt entry (name), gene name, corresponding protein description and sequence, sequence length, number (sequence) of peptides identified, species, log2-transformed protein intensities, GO annotation and keywords

are given. Ranking is done alphabetically according to protein names.
(XLSX)

**S3 Table. List of 4,666 human proteins quantified with a minimum of three valid values in at least one setup of the dual proteome shotgun samples of *S.* Typhimurium infected human or control HeLa cells by means of data-dependent acquisition (DDA).** Comparisons were analysed by t-testing and significant hits determined using as cut-off values a permutation based false discovery rate (FDR) of 0.01 and 0.001 (100 permutations) and a background variance parameter S0 of 0.1. Table headers are as in S1 Table.
(XLSX)

**S4 Table. List of 479 *S.* Typhimurium proteins quantified with a minimum of three valid values in at least one setup of the dual proteome shotgun samples of *S.* Typhimurium infected human HeLa cells by means of data-dependent acquisition (DDA).** Comparisons were analysed by t-testing and significant hits determined using as cut-off values a permutation based false discovery rate (FDR) of 0.05 and 0.01 (100 permutations) and a background variance parameter S0 of 1. Table headers are as in S1 Table.
(XLSX)

**S5 Table. List of 5,818 human proteins quantified with a minimum of three valid values in at least one setup of the dual proteome shotgun samples of *S.* Typhimurium infected human or control HeLa cells by means of data-independent acquisition (DIA).** Comparisons were analysed by t-testing and significant hits determined using as cut-off values a permutation based false discovery rate (FDR) of 0.01 and 0.001 (100 permutations) and a background variance parameter S0 of 0.1. (Regulated) host proteins previously found associated with *Salmonella*-modified membranes (SMM) [52] are indicated in indicated in columns 'SMM proteome (PMID: 25348832)' and/or 'ANOVA regulated SMM protein', respectively. Remaining table headers are as in S2 Table and cluster indication is as in S6 Fig.
(XLSX)

**S6 Table. List of 844 *S.* Typhimurium proteins quantified with a minimum of three valid values in at least one setup of the dual proteome shotgun samples of *S.* Typhimurium infected human HeLa cells by means of data-independent acquisition (DIA).** Comparisons were analysed by t-testing and significant hits determined using as cut-off values a permutation based false discovery rate (FDR) of 0.05 and 0.01 (100 permutations) and a background variance parameter S0 of 1. Table headers are as in S2 Table with regulon and genome locus and strand info added.
(XLSX)

**S7 Table. Significantly regulated (FDR $\leq$ 0.02) normalized human annotation enrichment scores across all 6 conditions (averaged expression values) assayed by data-independent acquisition (DIA).** Term enrichment was determined using the 1D annotation enrichment algorithm embedded in the Perseus software suite and p-values were corrected for multiple hypotheses testing using the Benjamini and Hochberg false discovery rate.
(XLSX)

**S8 Table. Regulated host proteins associated with *Salmonella*-modified membranes, implicated in transport, membrane or cytoskeleton organization, or representing GTPases.** From the 84 regulated SMM proteins identified in total according to [52], 29 ANOVA regulated host proteins associated with *Salmonella*-modified membranes, implicated in transport, membrane or cytoskeleton organization, or representing GTPases (selected from keywords)

are listed. Table headers are as in S2 Table and cluster indication as in S6 Fig.
(XLSX)

## Acknowledgments

We would like to acknowledge An Staes for her help performing LC-MS sample analyses and thank Prof. Jörg Vogel and Prof. Alexander Westermann for providing us the GFP-expressing *Salmonella* SL1344 strain JVS-3858 (*Ptet::gfp*) [22].

## Author Contributions

**Conceptualization:** Kris Gevaert, Petra Van Damme.

**Data curation:** Patrick Willems, Petra Van Damme.

**Formal analysis:** Patrick Willems, Margaux De Meyer, Petra Van Damme.

**Funding acquisition:** Kris Gevaert, Petra Van Damme.

**Investigation:** Ursula Fels, Patrick Willems, Petra Van Damme.

**Methodology:** Petra Van Damme.

**Project administration:** Petra Van Damme.

**Resources:** Petra Van Damme.

**Software:** Patrick Willems.

**Supervision:** Kris Gevaert, Petra Van Damme.

**Visualization:** Petra Van Damme.

**Writing – original draft:** Margaux De Meyer, Petra Van Damme.

**Writing – review & editing:** Ursula Fels, Patrick Willems, Margaux De Meyer, Kris Gevaert, Petra Van Damme.

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
