## [Decision Letter · Decision Letter 0]

6 Mar 2023

Dear Dr Van Damme,

Thank you very much for submitting your manuscript "Shift in vacuolar to cytosolic regime of infecting Salmonella from a dual proteome perspective" for consideration at PLOS Pathogens. As with all papers reviewed by the journal, your manuscript was reviewed by members of the editorial board and by several independent reviewers. In light of the reviews (below this email), we would like to invite the resubmission of a significantly-revised version that takes into account the reviewers' comments.

We cannot make any decision about publication until we have seen the revised manuscript and your response to the reviewers' comments. Your revised manuscript is also likely to be sent to reviewers for further evaluation.

Sincerely,

Andreas J Baumler

Academic Editor

PLOS Pathogens

Karla Satchell

Section Editor

PLOS Pathogens

Kasturi Haldar

Editor-in-Chief

PLOS Pathogens

orcid.org/0000-0001-5065-158X

Michael Malim

Editor-in-Chief

PLOS Pathogens

orcid.org/0000-0002-7699-2064

Reviewer's Responses to Questions

**Part I - Summary**

Reviewer #1: Intracellular pathogens including Salmonella can construct a special niche in host cells, by using virulence factors and some other mechanisms, to shelter from humoral immunity and proliferate, whereas host cells also deploy unique immune systems to combat that intracellularly invaded pathogens. Such interaction between intracellular pathogens and host cells are quite complicated and not easy to gain insight. In the manuscript entitled “Shift in vacuolar to cytosolic regime of infecting Salmonella from a dual proteome perspective” by Fels U. et al. attempted to dissect translated proteins both in host cell (HeLa cell) and Salmonella Typhimurium over course of bacterial infection by a new proteomics approach called “dual proteome profiling”. With using the approach, they found DIA mode is better than DDA one to detect low expressed peptides in Salmonella and could show trajectory of their expressions during infection, also patterns of some expressed proteins were consistent with previous reports. New proteomic approach that the authors established previously is potentially useful to analyzed proteins of both host and bacteria without enrichment, however, this version of manuscript has some serious issues. The proteomic approach remains incomplete, and results obtained by the analysis look not bringing new findings in this manuscript. Here, I would like to list up more detail. Addressing the questions and changing manuscript would be better suited for the publication.

Reviewer #2: Novelty:

- this is well-studied system so the novelty lies in the quality and depth of the analysis, and the conclusions that these deep measurements allow them to draw

Strengths:

- very strong technical approach to studying an important host/pathogen system

- very well executed

- appropriate infection model (MOI, time course, gentimycin addition, etc)

- some appropriate analyses of their dat and comparisons to others

Areas to improve:

- I would suggest some more comparisons to or mentions of some of the other proteomic studies that have been done on this system. I don't care so much about comparisons to transcriptomic studies - those typically aren't so relevant for understanding what happens at the proteome level anyway.

- one of the key features of Salmonella's infection system is its ability to establish the intracellular niche that it depends on to be able to replicate within the host cell. There is some analysis related to potential factors involved in surviving in the cytosol (lines 304 and onward) but mostly Salmonella needs to take over internal membrane systems (the Salmonella-containing vacuole and filaments). As I was reading the manuscript, I was hoping to learn something about how there might be changes in proteins involved in those processes. The data presented in Fig. 5 starts to go in that direction but only from the Salmonella side of the equation.

Reviewer #3: Here the authors present a dual proteomics time course of Salmonella Typhimurium infecting Hela cells. The proteome profiles of the pathogen and the host cell are analyzed after 2h, 4h, 8h, 16h and 24h. The authors also compare two proteomics approaches, DIA and DDA reporting that DIA is outperforming DDA both in terms of identifications and quantifications, especially for the bacterial pathogen, for DDA data, correlation of quantifications was shown to be higher than for DIA data. The authors present diagrams of the protein expression of different proteins of the host or the pathogen (significantly regulated ones) as well as functions and regulators of the host cell or S. typhimurium regulons.

**Part II – Major Issues: Key Experiments Required for Acceptance**

Reviewer #1: 1. The new approach “dual proteome profiling” without a priori bacteria enrichment in this study sounds attractive but remains incomplete. Salmonella proteome contains around 4,500 proteins but this approach detected only less than 900. Previous reported proteomic research demonstrated detection of Salmonella proteins around 1,800 with enrichment infected cells (Li et al, mSystems, 2019; RMID:30984873), suggesting “dual proteome profiling” is difficult to be taken the place of previous approach without further modification.

2. Human sample contains a control (noninfected control), but Salmonella sample doesn’t contain any proper control. Expression of bacterial proteins are quickly changed by response to environment. I think “2 hpi” is too late to see early time responses and compare with the responses at later time points, thus “0 hpi” sample for bacterial side should be needed. Furthermore, in Salmonella proteomic results, I think new findings seem not to be presented in the manuscript.

3. The author should prepare graphs in figures representing that they want to say. In some places, authors take us to see supplementary tables, but such tables are just lists of protein expression data. For example, “….and unique number of peptides identified (Supplementary Tables S1A-B)” on line 158 of the manuscript, I couldn’t figure out “unique number of peptides” on the tables. “…significant upregulation of metal ion transporters and… (Supplementary Table S2D)“ on line 291, representing by graphs should be better. In addition of that, some figures in the manuscript need to be modified. Fig. 1 should add PCA of Salmonella in the over course of infection as well as host cells. Individual protein expression patterns regulated by the regulators (e.g., hilA, phoPQ, or others) in Fig. 5 should be shown. I also don’t know why Fig. 6 is involved in Discussion. It should be moved to Result section.

4. Some biological experimental designs are not good. Authors show intracellular bacteria cfu (in well? per mL?) and cytotoxicity induced by infection with Salmonella in Fig. S1, it should include results at later time points (16 hpi and 24 hpi). Intracellular cfu of a prgH mutant stain should be shown in Fig. S1A as well. In the LDH cytotoxicity assays, “Target Cell Maximum LDH Release Control” is required (see the instruction), otherwise it can’t be rule out that possibility of “non-Infected” control cells are also dying.

Reviewer #2: I do not feel that there are additional wet experiments required. Rather, the following bioinformatic analyses could enrich the manuscript:

- comparison to other Salmonella proteomic studies - similarities/differences in classes of proteins expressed

- analysis of host membrane trafficking proteins altered in their study

Reviewer #3: The work seems to be well done and is clearly written. The authors present a more comprehensive proteomics analyses and identify more proteins than published before, however, my main problem is that I did not see what the new findings are, or they were not presented well. All results seem to be confirmatory (references are given) and I am missing that a specific, new result is highlighted. Thus, I recommend to better define the novelty of the results obtained by this dual proteomics approach and to better discuss how the host cell and the pathogen interact, react to each other.

**Part III – Minor Issues: Editorial and Data Presentation Modifications**

Reviewer #1: (No Response)

Reviewer #2: Could not decide whether the above "Major" issues should be Minor, instead

Reviewer #3: Line 34 it reads “bacterial proteome response during infection progression infection by permitting quantification of low » please delete one “infection” it is double

Lines 304-308 it reads ….” Since extensive transcriptional reprogramming was shown to accompany the successful colonization of the epithelial cytosol, a niche occupied by a hyper-replicating Salmonella sub-population especially at later times post-infection, it is noteworthy that upregulation of cytosol signature genes previously reported to be transcriptionally upregulated in Gram-negative pathogens colonizing the cytosol were also observed as being regulated in our proteome study »

How did you control whether bacteria where cytosolic or vacuolar? Did you follow GFP bacteria before protein preparation or is this an assumption form the protein signature?

Figure 2: could perhaps be in the supplement, it is again very general, and no real functional conclusion can be drawn from this figure

Figure 3: what are the different clusters and numbers e.g. Cluster 1 (#286). In the way it is shown this figure is very general and one does not get the functional implications.

I would like to see figures S4 and S5 in the main text

PLOS authors have the option to publish the peer review history of their article (what does this mean?). If published, this will include your full peer review and any attached files.

Reviewer #1: No

Reviewer #2: **Yes: **Leonard Foster

Reviewer #3: No
---

## [Editor Report · Decision Letter 1]

19 Jun 2023

Dear Dr Van Damme,

We are pleased to inform you that your manuscript 'Shift in vacuolar to cytosolic regime of infecting Salmonella from a dual proteome perspective' has been provisionally accepted for publication in PLOS Pathogens.

Best regards,

Andreas J Baumler

Academic Editor

PLOS Pathogens

Karla Satchell

Section Editor

PLOS Pathogens

Kasturi Haldar

Editor-in-Chief

PLOS Pathogens

orcid.org/0000-0001-5065-158X

Michael Malim

Editor-in-Chief

PLOS Pathogens

orcid.org/0000-0002-7699-2064
---

## [Editor Report · Acceptance letter]

31 Jul 2023

Dear Dr Van Damme,

We are delighted to inform you that your manuscript, "Shift in vacuolar to cytosolic regime of infecting Salmonella from a dual proteome perspective," has been formally accepted for publication in PLOS Pathogens.

Best regards,

Kasturi Haldar

Editor-in-Chief

PLOS Pathogens

orcid.org/0000-0001-5065-158X

Michael Malim

Editor-in-Chief

PLOS Pathogens

orcid.org/0000-0002-7699-2064